# Point-of-care testing of hyponatremia and hypernatremia levels: An optoplasmonic biosensing approach

Abdullah Mohammad Tanvirul Hoque[1☯], Abrar Islam [2,3☯]*, Firoz Haider[4],
Rifat Ahmmed Aoni[5], Rajib Ahmed[6,7]

1 The Wyant College of Optical Sciences, The University of Arizona, Tucson, Arizona, United States of America, 2 Department of Electrical and Electronic Engineering, Bangabandhu Sheikh Mujibur Rahman Science and Technology University, Gopalganj, Bangladesh, 3 Department of Electrical Engineering, The Pennsylvania State University, University Park, Pennsylvania, United States of America, 4 Faculty of Science & Technology, Charles Darwin University, Darwin, Northern Territory, Australia, 5 The Mork Family Department of Chemical Engineering and Materials Science, University of Southern California, Los Angeles, California, United States of America, 6 School of Medicine, Stanford University, Palo Alto, California, United States of America, 7 School of Bioengineering and Health, Wuhan Textile University, Wuhan, Hubei, China

☯ These authors contributed equally to this work.
* abrar@psu.edu

## Abstract

In the human body, sodium ion (Na+) is the principal extracellular cation and its imbalance in blood or lymph results in either hyponatremia or hypernatremia conditions. Any of these scenarios have fatal effects on a patient's health over time. However, the existing technologies limit quick point-of-care (POC) testing options. To overcome the challenges associated with prompt diagnosis, we proposed a photonic crystal fiber (PCF) based surface plasmon resonance (SPR) sensor for in situ rapid testing of Na+ ion levels in patient's blood samples. A unique PCF sensor with a modified square lattice is engineered to achieve a propagation-controlled core that permits enhanced control over wave guidance resulting in improved sensing performance. Plasmonically active conducting indium tin oxide (ITO) is used to omit the limitations that arise from undesired oxidation with time and to enable higher spectral tunability. The finite element method (FEM) is used to perform the numerical analysis on performance and all sensor parameters are optimized to ensure the best sensing outcome. The structural asymmetry-induced birefringence enforced a superior y-polarized sensing response compared to that of the x-polarized mode for the proposed sensor, where it proffers the highest wavelength sensitivity of 15,157 nm/RIU with a detection resolution of $6.61 \times 10^{-6}$ RIU. Additionally, the sensor exhibits a maximum amplitude sensitivity of 470 RIU$^{-1}$ and an outstanding figure of merit (FOM) of 171 RIU$^{-1}$. Due to the high sensitivity and resolution, we infer that the proposed sensor will be a promising POC alternative to the conventional serum electrolyte panels used in present medical practice.

**Data availability statement:** Data underlying the results presented in this paper are available from the COMSOL file attached as the Supporting information.

**Funding:** The author(s) received no specific funding for this work.

**Competing interests:** The authors have declared that no competing interests exist.

## Introduction

INSIDE a human blood circulatory system, sodium ion (Na$^+$) is one of the most crucial electrolytes of extracellular fluid which is responsible for the generation of action potentials in nervous and cardiovascular tissues [1]. This makes the balanced presence of Na$^+$ an imminent requirement for the body to function properly. Although our body possesses a sophisticated mechanism for maintaining the optimal amount of electrolytes within our bloodstream, reported cases of unstable sodium levels are very frequent in patients with kidney problems, different viral & microbial infections, diarrhea & diabetes mellitus [2,3]. Cancer patients are sometimes required to be under constant observation for abnormal serum sodium level [4]. Repetitive encounters of out-of-balance electrolytes in bodily fluid are found in COVID-19 patients which led sodium concentration in blood being an indirect prognostic marker of the disease [5]. Among patients admitted to hospitals, individuals with electrolyte imbalance had a greater mortality ratio than the rest [6]. Lately, Lo et al. [7] suggested dysnatremia or extreme abnormality of plasma Na$^+$ level is the highest reported electrolyte disorder in patients with liver cirrhosis and heart failure.

Since normal serum Na$^+$ concentration is mostly recognized within the range of 135–145 mmol/L (135–145 mEq/L), hyponatremia (i.e., low blood sodium level) is defined as a concentration of sodium being less than 135 mmol/L [8]. On the other hand, hypernatremia is typically characterized as a serum sodium level greater than 145 mmol/L [9]. The imbalance of bodily sodium can directly be associated to any malfunctioning of kidney, restricted operations of metabolism and circulatory systems and deficiencies of the overall immune system [10]. An abrupt change of the Na$^+$ level within mammals' internally results in cellular swelling or shrinkage due to imbalanced osmotic pressure. Brain cells are heavily prone to this irregularity and can result in chronic subdural hemorrhage, permanent coma or even death in worst case scenarios [11]. Additionally, these patients typically have a high possibility of having cerebral edema (in hyponatremia) or vascular rupture (in hypernatremia) [12]. Even a mild Na$^+$ ion imbalance typically causes dizziness, brain swelling, headaches, nausea, excessive blood pressure, and sudden loss of stamina [13]. Moreover, electrolyte measurements are important in the differential diagnosis of a variety of illnesses, including diabetic ketoacidosis and renal tubular acidosis [14]. Rigorous physical exams and patient history are necessary to help identify the core reasons of sodium imbalance and to administer any treatments [8]. In certain situations like individuals with persistent hyperosmolality, intensive treatment with hypotonic fluids may be required immediately [15].

The conventional way of testing blood ion levels is the serum electrolyte test, which is not in situ since it involves sample centrifugation in the laboratory. Following that, the serum is placed into sample cups for examination on a fully automated clinical chemistry analyzer. Due to the delay between centrifugation and sample analysis, elevated false positive results are common findings in the laboratory [16]. For further validation, patients oftentimes need to undergo a number of pricey procedures, including liver function tests, computed tomography (CT) scans, and electrocardiography [17,18]. Among point-of-care (POC) approaches, arterial blood gas analyzers and auto-analyzers proffered consistent results for the measurement of potassium ion, whereas significant discrepancies are reported by Jain et al. [19] in the case of sodium. A recent gas analyzer i-Smart 300E was able to demonstrate decent operation but in conjunction with a bulky setup involving disposable cartridge, reagents, waste bags, tubing, and sample probes [14]. Therefore, a fast and affordable compact POC testing system of blood electrolytes is critically needed in current medical practice to expedite the diagnosis without compromising accuracy and repeatability.

Here, we propose a novel bio-optical method for real-time measurement and monitoring of Na$^+$ levels in serum, which can be employable at both medical and patient's home

settings. In this work, we leveraged the change in optical properties (refractive index) of serum due to any minute variation in the sodium ion concentration in blood and engineered a lossy waveguide (photonic crystal fiber or PCF) to quantify the change. The amplitude of the loss in optical waveguides and the spectral position of the loss peaks are very susceptible to the refractive index of surrounding materials when the whole structure is plasmonically active. The plasmonic microstructure is easy to obtain as any conductor-dielectric interface is physically bound to generate surface plasmon polariton (SPP) under an applied electric field through electromagnetic interactions. The energy required for the oscillation of SPPs is collected from the applied light into the waveguide rendering the propagation loss in principle. When the frequency of the transmitted electromagnetic wave matches the free electron oscillation frequency, the loss accentuates which is defined as the surface plasmon resonance (SPR). That implies, at resonance, the confinement of light inside the waveguide is minimal and the loss profile at this resonant wavelength is the basis of our measurement scheme.

Using the SPR phenomena, refractive index (RI) sensors are positively advocated by researchers and sensor technologists because of their highly sensitive nature, design flexibility, and compactness [20, 21]. To leverage these favorable features, metal-insulator-metal structures and meta-absorber systems are being investigated recently [22–24]. However, the popular choices of SPR sensor platforms are still either prism-based or fiber-based. Prism-based sensors were a traditional approach in SPR technologies, which are large and heavy in design since optomechanical rotary parts are involved [25]. On the other hand, miniaturization and portability are attained when metal-coated optical fibers are used instead of prisms. However, precise control over the doping concentrations in graded-index fibers and the jacket removal procedure make the fabrication of conventional fiber-based sensors relatively complicated. These reasons favor PCF as the leaky waveguide over other candidates for sensor applications, since controlled confinement of the propagating light and SPP excitation can be done very conveniently.

Here, a novel PCF-based SPR sensor is designed and critically investigated in terms of realization feasibility and sensing performance as an in-situ blood $Na^+$ testing kit. There are reported numerical studies on SPR sensors for label-free general-purpose refractometry of different chemical and biomolecular solutions with moderate sensitivities [26]. For example, the highest wavelength sensitivity of a proposed liquid sensor went up to 10,000 nm/RIU using a titanium nitride-coated fiber [27]. More recently, plasmonic sensors targeting specific bio-analytes received higher attention by researchers owing to the drive of medical instrumentation industries. In 2022, a biosensor by Rajeswari and Revathi [28] yielded a maximum sensitivity of 6701.03 nm/RIU aiming for early screening of cancerous cells. Another machine learning-assisted SPR-based malaria sensor exhibited a sensitivity of 12,142 nm/RIU, 9736 nm/RIU and 8241 nm/RIU for ring, trophozoite and schizont stages of malaria parasite reproduction cycle, respectively [29]. Lately, a photonic COVID sensor proffered a spectral sensitivity of 3948 nm/RIU [30]. In 2024, a hemoglobin sensor developed by Dai et al. [31] was able to achieve wavelength sensitivities up to 7500 nm/RIU. Our proposed sensor surpassed these recently investigated biosensors in terms of sensing performance as well as has shown merit in outperforming typical bioassays for serum electrolyte concentration analysis. Transportability as an integrated POC kit and fast turnaround with high repeatability are the most promising aspects of our design [32]. In addition, this study explored a completely new avenue for estimating bodily present $Na^+$ concentration in POC scenarios utilizing biophotonic engineering. Our finite element method (FEM)-based numerical analysis validates that the sensor has good fabrication tolerances of structural parameters to achieve the best possible sensing operation.

## Sensor design and theory

In the sensor, a thin film of indium tin oxide (ITO) is used as a plasmonic layer due to its enhanced freedom of optical tunability by regulating the oxygen content. Unlike silver and gold, ITO does not suffer from oxidation problems in aqueous environments and exhibits an operational window around telecommunication wavelengths. One common challenge with PCF-based plasmonic sensors is to meticulously design the leakage paths of the light energy to excite the SPPs when there are thick cladding layers. Scaling the core air holes down to optimal sizes enables unprecedented control over the magnitude and direction of this energy flow [33]. Our POC-compatible biosensor is based on a diligently modified semi-square 2-ring lattice structure which ensures very good light-tailoring inside the PCF as we desired. This is the first demonstration of a propagation controlled core [34] being implemented with a square-shaped lattice by breaking the hexagonal periodicity. Additionally, instead of drilling, this sort of fiber structure is easy to fabricate by stacking commercially available micro-capillaries and drawing the preform [35]. The four-fold asymmetry of the lattice mandates polarization dependency for our sensor resulting in a highly birefringent propagation profile. Because of the high flexibility offered by available PCF manufacturing technologies, the plausible ways of designing SPR sensors are vastly exploited [26]. However, internal infusion of analyte into small capillaries is frequently criticized for posing uncontrollable loss [36]. Some reported works studying these sorts of SPR sensor configurations can be found in ref [37–39]. In addition, surface roughness arisen from fiber polishing is highly undesirable for any kind of actuation applications. For these reasons, we adopted an external sensing approach employing an unpolished ITO-coated PCF. This means the serum sample can be glided onto the functionalized sensitive interface of the device from outside, which helps fast and complication-free cleaning and reusing.

Fig 1a and 1b depict the cross-sectional view of the sensor and the stacked capillary arrangement of the vertically elongated core PCF, respectively. The air holes radius at the core of the fiber is scaled down to regulate the light confinement and open energy leakage paths opposite to each other. Four solid glass rods are inserted adjacent to the center to stabilize the stacked microstructure (see Fig 1b). This configuration allows better SPP excitation by rerouting the evanescent electric fields to four specific locations which resulted in increased analyte-ligand interactions. In this design, fused silica is used as the fiber material, and large air holes are introduced as the cladding so that total internal reflection limits the complete dissipation of the guided photon energy. Sellmeier equation is used to model the optical index of the background silica for the infrared (IR) band [40];

$$n^2(\lambda) = 1 + \frac{B_1\lambda^2}{\lambda^2 - C_1} + \frac{B_2\lambda^2}{\lambda^2 - C_2} + \frac{B_3\lambda^2}{\lambda^2 - C_3} \tag{1}$$

Here, $n$ is in the RI of fused silica and $\lambda$ is the associated wavelength in the micrometer unit. Sellmeier constants are collected and plugged into the Equation (1) from [41] where $B_1 = 0.69616300$, $B_2 = 0.407942600$, $B_3 = 0.897479400$, $C_1 = 0.00467914826$, $C_2 = 0.0135120631$, $C_3 = 97.9340025$. On top of the PCF, the conducting ITO layer is placed which can be deposited either through atomic layer deposition or any other nanoparticle deposition technique. For simulation purpose, we used the empirical Drude model below to approximate the complex dielectric constant of ITO [42]:

$$\varepsilon(\omega) = \varepsilon - \frac{\omega_p^2}{\omega^2 + j\omega\Gamma} \tag{2}$$

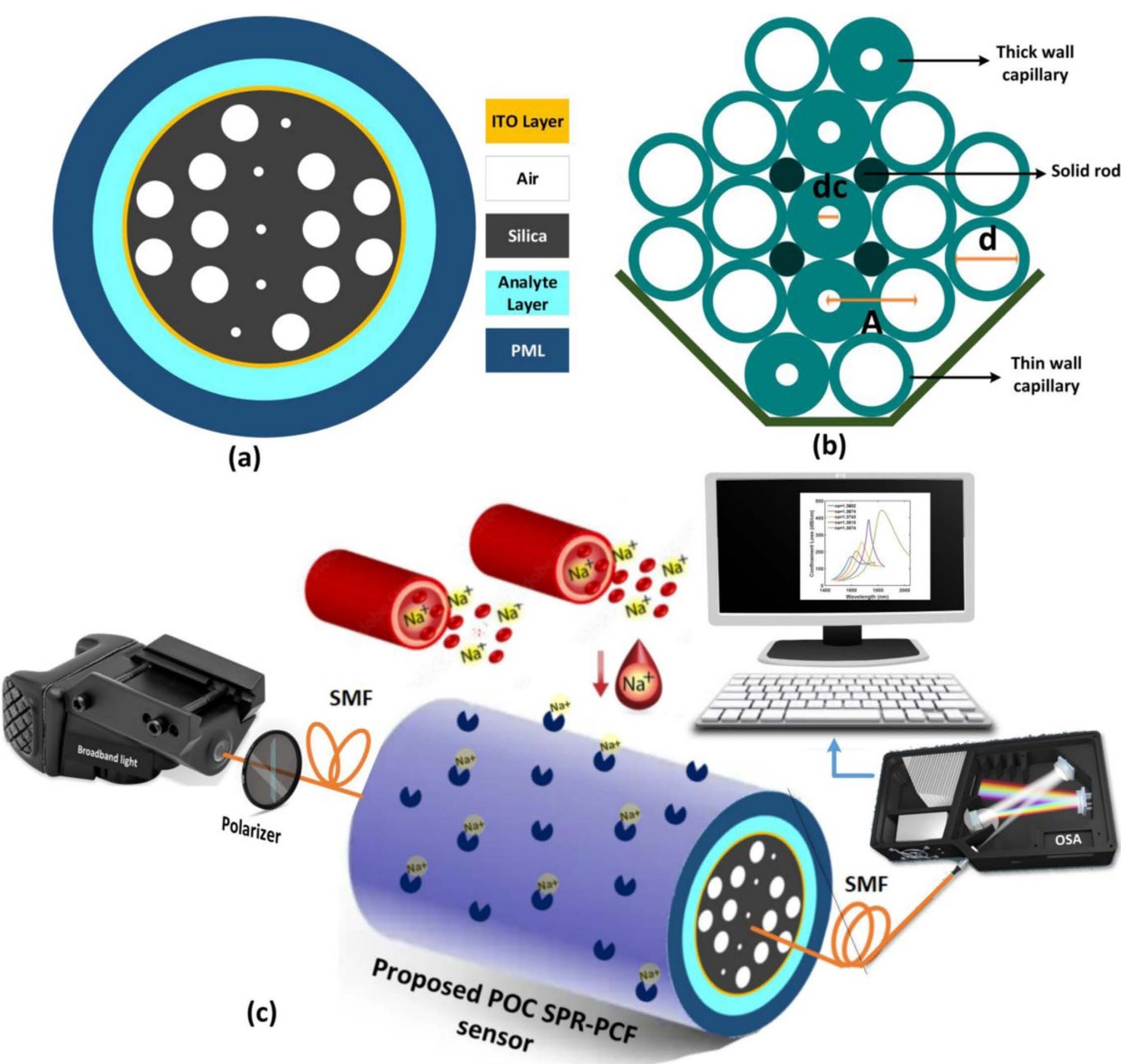

**Fig 1. (a) Cross-section of the proposed sensor, (b) Stacked PCF preform, (c) Schematic diagram of the experimental setup.**

Where, the intra-band dielectric constant, $\varepsilon = 3.964$; damping coefficient, $\Gamma = 0.111 eV$;

plasma frequency, $\omega_p^2 = \left(\dfrac{ne^2}{\mu\varepsilon_0}\right)^{\frac{1}{2}} = 2.19 eV$; provided that $\mu = 0.3$ times free electron mass. To

apply boundary conditions to our 2D cross-section, a cylindrical perfectly matched layer (2

μm thick) is employed for absorbing the scattered lights. Detailed optimization of dimensional parameters and computation of nanofabrication tolerances are articulated in the following section of this paper. Nevertheless, the optimal values are listed in Table 1.

To make our sensing scheme operational, we interpreted the molar density of serum sodium in terms of its RI and realized the immobilization of Na⁺ ion at the sensor surface.

**Table 1. Optimized sensor parameters.**

| Parameters | Value |
|---|---|
| Pitch size of PCF, Λ* | 2.2 μm |
| Cladding air hole diameter, d | 1.43 μm |
| Core air hole diameter, dc | 0.44 μm |
| Thin ITO layer thickness, t | 50 nm |

*Pitch size (Λ) is defined as the center-to-center spatial separation between two adjacent air holes.

At room temperature, the plasma sodium RI can be correctly approximated by Equation (3), where $C$ denotes concentration in mgdL$^{-1}$, and $k$ indicates the concentration factor [43];

$$n_{Na^+} = 1.3373 + 1.768 \times 10^{-3} \left( \frac{C \star k}{393} \right) - 5.8 \times 10^{-6} \left( \frac{C \star k}{393} \right)^2 \tag{3}$$

Table 2 summarizes several values defined by the relation given above and shows how different diagnostic decisions can be made based on the extracted data from our proposed system. We assumed a midpoint (139.1 mmol/L or 320 mg/dL) of the prescribed normal sodium concentration range to be the Na$^+$ level in a healthy body.

Carboxylate or sulfonate bifunctional ligand and calcium silicate hydrate gel pores can be alternately used for immobilizing Na$^+$ atoms over our predesignated plasmonic surface [44, 45]. Then, the attached ions will correspond to a certain RI for the analyte layer and modulate the resonant wavelengths of our fiber sensor. Fig 1c depicts the functional schematic of our biosensing setup. Initially, a broadband light source followed by a polarizer inputs the IR wavelengths into the fiber over a preassigned time interval (typically a few seconds). Due to the leaky waveguide design, the SPPs bonded with sodium particles through ligands receive energy from the propagating light. Therefore, distinguishable confinement loss peaks appear when SPR occurs at specific wavelengths for individual sodium plasma samples if the concentration (or RI) is different in each case. The scanned spectral data is fed into a computer after being collected by an optical spectra analyzer or charged coupled device. Lastly, a quick post-processing and a comparison with modeled reference loss profiles give out the final results. Furthermore, this arrangement can be miniaturized into a simple and reusable hand-held POC device with a diode laser, mini-spectrometer and a microcontroller since a sensing interface of a few millimeters is adequate for highly accurate measurements.

**Table 2. The RI depicted by patient's plasma for different sodium levels.**

| Analyte | Concentration* | | RI* | Remarks* |
|---|---|---|---|---|
| | mmolL$^{-1}$ | mgdL$^{-1}$ | | |
| Na$^+$, $k = 30$ | 86.95 | 200 | 1.3602 | Extreme hypo |
| | 113.00 | 260 | 1.3674 | Hyponatremia |
| | 139.10 | 320 | 1.3743 | Normal |
| | 165.20 | 380 | 1.3810 | Hypernatremia |
| | 191.30 | 440 | 1.3874 | Extreme hyper |

*The operation range and remarks are not limited to these 5 concentrations only. Experimentation with these 5 concentration levels is done as the proof-of-concept. The sensor is operational for all RIs ranging 1.33–1.40.

## Results and discussions

To justify our proposed detection model, we showed the FEM-based analysis results with COMSOL Multiphysics, where the entire geometry is meshed into 15046 domain elements. Fig 2 shows the electric field distribution throughout the whole cross-sectional area of this sensor at normal sodium level for y-polarized light input under fulfilled resonance conditions. Our polarization-dependent geometry resulted in a prevalent y-polarization excitation response in reference to the x-polarization mode operation. Electric field intensity profiles in fundamental core-guided mode (Fig 2a) and SPP mode (Fig 2b) indicate strong coupling between them. In addition, the energy leakage pathways originated from our engineered propagation-controlled core are revealed from the intensity color map. The phase matching (at resonance, real RI of core mode = real RI of SPP mode) of the aforementioned fiber modes is shown in Fig 2c along with the sharp modal loss that appeared at the resonant wavelength of 1677 nm. The fiber propagation loss is evaluated as follows [46];

$$\alpha\left(\frac{dB}{cm}\right) = 8.686 \times \frac{2\pi}{\lambda(\mu m)} I_m\left(n_{eff}\right) \times 10^4 \tag{4}$$

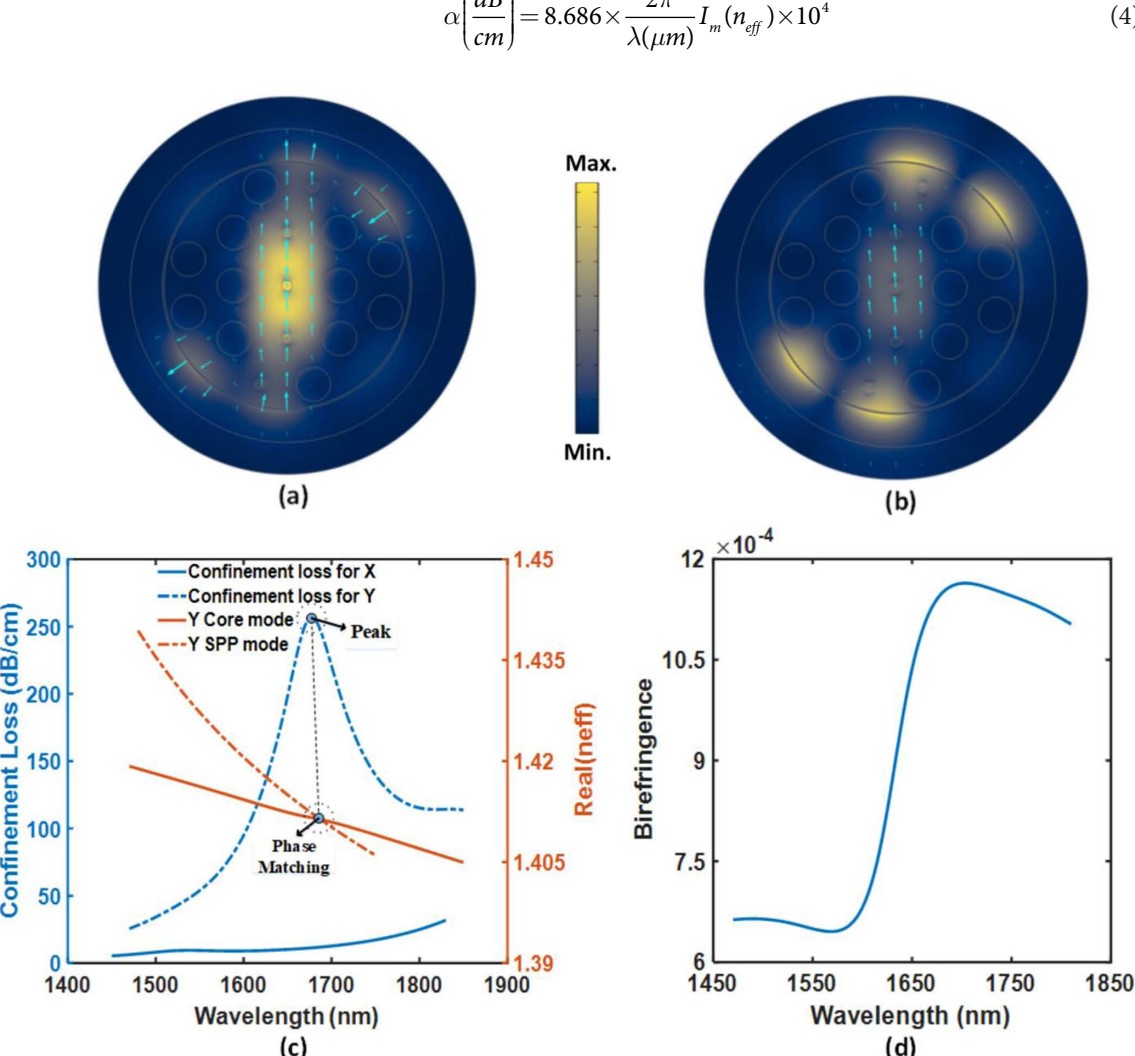

**Fig 2. (a)** Electric field profiles of the core-guided mode and **(b)** SPP mode at analyte RI 1.3743. **(c)** Satisfied phase-matching condition, and **(d)** Birefringence behavior.

Here, $\lambda$ = applied wavelength & $I_m$ = extinction coefficient of the modal effective index. When a light guiding system responds differently for two orthogonally linear polarized lights, the difference in their real RIs is defined as the birefringence profile of that system. The birefringence spectrum of the proposed photonic sensor for analyte RI 1.3743 is shown in Fig 2d. We can see intensified birefringent behaviors (up to $11.4 \times 10^{-4}$ RIU) of the structure in longer wavelengths owing to increased light penetration depth at that region [47]. Any asymmetrical structure has a natural tendency to exhibit strong birefringence which enables better control of the polarization states of the input signal and increases the stability of the optical device [30].

We can see, both resonant wavelength and peak loss magnitude vary depending on the change in analyte layer RI. This is because, due to variations in analyte concentration, the net number of sodium ions varies and attaches to the carboxylate/sulfonate ligands accordingly. The number of active ligand-Na$^+$ conjugates is dictated by analyte RI and thus, the required energy changes for satisfied SPR conditions for different analyte samples. The shifts in resonant wavelengths could be blue or red with an increase of ion concentration based on the type of PCF used. For our particular case, we observed redshifts of SPRs as the sodium level goes up in the patient's blood stream. Fig 3a portrays this fact in our working wavelength region for serum sodium concentrations tabulated in Table 2. The molar concentration of 86.95 mmol/L depicts the lowest loss of 171 dB/cm at 1599 nm, whereas the highest loss of 444 dB/cm is evident for 119.3 mmol/L Na$^+$ level at 1826 nm. All SPR data points of interest from those loss curves are summarized in Table 3. Here, it is noteworthy that the forward-shifting of resonance is also causing increased loss depths. This is because of a subtle continuous decline in RI contrast difference between the fundamental mode and the plasmon mode for escalated concentration of Na$^+$ ions [48].

The SPR wavelength shift per unit RI variation of the analyte is specified as the wavelength or spectral sensitivity ($S_\lambda (nm/RIU) = \Delta\lambda_{res}/\Delta RI_{analyte}$). The deeper light penetration into ITO as the working wavelength pushes towards mid-IR results in improved sensitivity in spectral interrogation. The highest wavelength sensitivity proffered by our presented biosensor is 15,157 nm/RIU at RI 1.3810 and the lowest magnitude is found 5,000 nm/RIU at RI 1.3602. Instead of analyzing the whole effective wavelength range, the amplitude-based

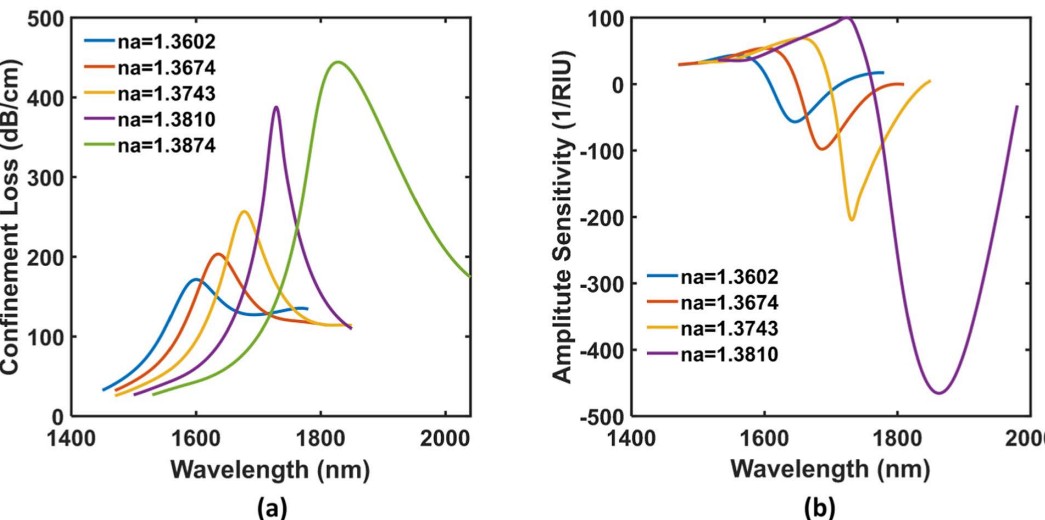

**Fig 3. (a) Confinement loss curves and (b) Amplitude Sensitivity curves of the proposed sensor.**

**Table 3. Performance analysis for the proposed PCF sensor in the detection of hyponatremia and hypernatremia.**

| Analyte RI | Res. Wave. (nm) | Peak Loss (dB/cm) | Res. Shift (nm) | Wave. Sens. (nm/RIU) | Ampl. Sens. (RIU$^{-1}$) | FWHM (nm) | FOM (RIU$^{-1}$) | Wave. Res. (RIU) | SNR | DL (RIU²/nm) |
|---|---|---|---|---|---|---|---|---|---|---|
| 1.3602 | 1599 | 171 | 36 | 5000 | -58 | 196 | 26 | $2.00 \times 10^{-5}$ | 0.184 | $1.32 \times 10^{-9}$ |
| 1.3674 | 1635 | 203 | 42 | 6087 | -99 | 167 | 37 | $1.64 \times 10^{-5}$ | 0.251 | $1.08 \times 10^{-9}$ |
| 1.3743 | 1677 | 257 | 52 | 7762 | -205 | 143 | 55 | $1.29 \times 10^{-5}$ | 0.364 | $8.85 \times 10^{-10}$ |
| 1.3810 | 1729 | 388 | 97 | 15157 | -470 | 89 | 171 | $6.61 \times 10^{-6}$ | 1.090 | $4.40 \times 10^{-10}$ |
| 1.3874 | 1826 | 444 | N/A | N/A | N/A | 229 | N/A | N/A | N/A | N/A |

investigation method for sensitivity characterization is also used here for challenging our capability under different detection approaches. Amplitude sensitivity is mathematically addressed as follows [49],

$$S_A\left[RIU^{-1}\right] = -\frac{1}{\alpha(\lambda, RI_{analyte})}\frac{\partial\alpha(\lambda, RI_{analyte})}{\lambda, RI_{analyte}} \tag{5}$$

Fig 3b displays the amplitude sensitivities for the listed RIs around their maximal values. Akin to $S_\lambda$, similar physics lies behind the successively improved $S_A$ at higher wavelengths. The sensor sensitivity peaks at test RI 1.3810 (concentration = 380 mg/dL) to a magnitude of 470 RIU$^{-1}$ in the amplitude interrogation as visualized in the figure. Sensor resolution is the measurement of the smallest quantifiable fluctuation in sample RI that can be detected or sensed by the device. The sensor resolution can be determined by the expression given below [48],

$$R(RIU) = \frac{\Delta RI_{analyte} \times \Delta\lambda_{min}}{\Delta\lambda_{res}} \tag{6}$$

Provided that the assumed minimal spectral resolution ($\Delta\lambda_{min}$) is 0.1 nm which is standard for SPR-based technologies. The highest spectral resolution is obtained at the same point where the sensitivity was maximized (for RI = 1.3810) which is $6.61 \times 10^{-6}$ RIU. These performance parameters are tabulated in Table 3 as well for convenience. Moreover, Table 4 gives a comparative impression on the performance of our work with recently reported SPR refractive index sensors for other bio-analyte and chemical detection applications.

Although a sensor's sensitivity can give a direct indication of the merit of detection and the degree of performance, there are additional factors to be considered for critically assessing a practicable sensor. Figure of merit (FOM) can be achieved as high as 171 RIU$^{-1}$ for this sensor which results from the steep resonance loss peaks (narrow full-width half maxima, FWHM) exhibited by the leaky PCF. Both FOM and FWHM are illustrated graphically in Fig 4a for our experimental sodium densities. The SPR wavelengths are fitted by a second order equation to

**Table 4. Performance comparison of the proposed sensor with recently reported sensor.**

| Ref. | Structure Type | RI Range | Pol. Mode | Max. Wave. Sens. | Max. Wave. Reso. | Max. Amp. Sens. |
|---|---|---|---|---|---|---|
| [20] | Titanium Nitride Coated PCF-SPR Sensor | 1.385–1.40 | Y-pol | 10,000 | $2.0 \times 10^{-5}$ | 70 |
| [21] | SPR-PCF Sensor For Early-Stage Cancer Detection | 1.3246–1.3634 | X-pol, Y-pol | 6701.03, 5154.63 | – | 110 |
| [22] | MXene-coated PCF for Malaria Detection in RBCs | 1.36–1.40 | X-pol | 12,142 | $1.21 \times 10^{-5}$ | – |
| [23] | PCF Biosensor for SARS-CoV-2 Particles | 1.3348–1.3576 | Y-pol | 3,948 | $2.53 \times 10^{-5}$ | 240 |
| [24] | U-channel SPR-PCF for Hemoglobin Sensing | 1.26–1.42 | Y-pol | 7,500 | $6.0 \times 10^{-3}$ | – |
| [28] | D-shaped PCF Sensor with Graphene and ZnO | 1.36–1.41 | Y-pol | 4,486 | $1.667 \times 10^{-5}$ | – |
| Proposed sensor | Testing of Hyponatremia and Hypernatremia Level Sensor | 1.3602–1.3874 | Y-pol | 15,157 | $2.0 \times 10^{-5}$ | 470 |

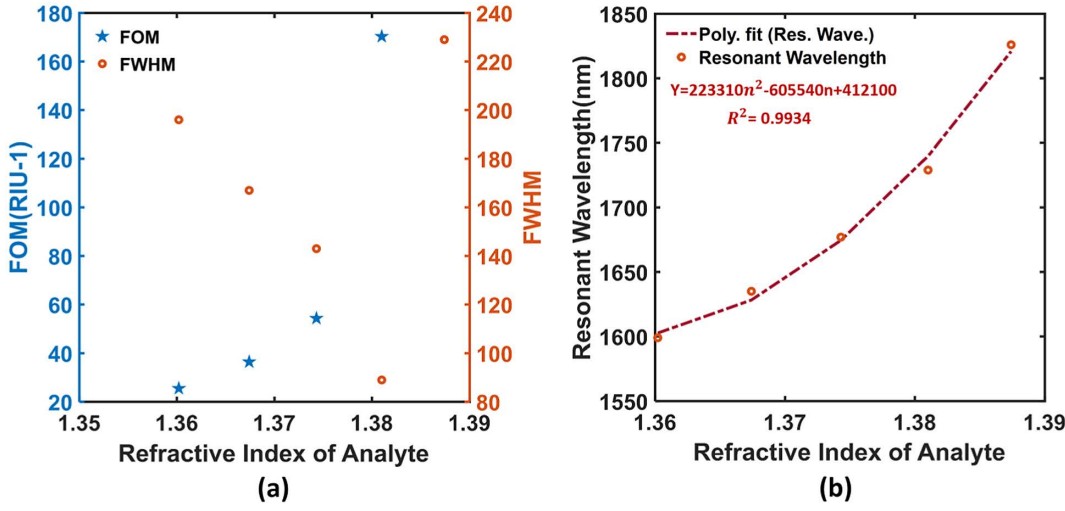

**Fig 4. (a) FOM & FWHM points and (b) Polynomial Fitting curve of the proposed sensor.**

observe the trend in sensing responses. The fitted curve is plotted against the analyte RIs in Fig 4b, where we can see very good $R^2$ (~unity) is found. The signal quality at resonant peaks and the minimum blood plasma sample required for viable operation can be determined by the signal-to-noise ratio (SNR) and limit of detection (DL) of the device. Our sensor demonstrated very promising SNRs and DLs for all of the cases as can be seen in the performance Table 3 (column 10–11). The proposed sensor exhibits the maximum SNR and DL values of 1.0899 and $1.32 \times 10^{-4}$ RIU²/nm, respectively. Using the following two equations, SNR and DL of the sensor have been evaluated.

$$SNR = \frac{\Delta\lambda_{res}}{FWHM} \tag{7}$$

$$DL = \frac{R_{min}}{S_\lambda} \tag{8}$$

## Optimization and tolerance analysis

Structural parameters play crucial roles in fine-tuning the wave-guidance which ultimately influence the final sensing performance. Therefore, to obtain sensitivity-intensified performance, we examined the impact of dimensional variations by taking into account different PCF parameters. Our optimization gives estimations on sensor tolerances for possible imperfections that might appear after fabrication. The light penetration into a thin film and the excitation of the immobilized analyte ions are governed by the optical property and thickness of the plasmonic layer as well as the efficiency of the designated leakage pathways. Therefore, to begin with, we investigated different variations of the ITO layer's depth (45 nm, 50 nm & 55 nm) at RIs 1.3674 and 1.3743, which are shown in Fig 5. During this successive thickening of ITO layer, at sample RI 1.3674, we got the peak losses of 167 dB/cm, 203 dB/cm, and 251 dB/cm, respectively for the aforementioned thicknesses. For the RI of 1.3743, the sensor also shows the peak losses of 200 dB/cm, 257 dB/cm, and 347 dB/cm following the same order (see Fig 5a). The wavelength sensitivity ($S_\lambda$) has improved as the ITO layer's thickness is increased which are 5072 nm/RIU, 6087 nm/RIU and 6957 nm/RIU due to the resonant wavelength

shifts of 35 nm, 42 nm and 48 nm respectively. Similarly, in the case of amplitude sensitivity ($S_A$), we got the peak values of 65 RIU$^{-1}$, 99 RIU$^{-1}$, and 152 RIU$^{-1}$ correspondingly (see Fig 5b). However, from the figure, it is evident that the values of the peak losses and $S_A$ are intensified with the increase of the depth of the plasmonic material. Here, we chose t = 50 nm over 45 nm and 55 nm for the proposed sensor because of the loss profile consideration for practicability. For t = 45 nm, though the sensor illustrates the lowest peak loss value, it has the lowest peak $S_A$ value. And for t = 55 nm, our sensor exhibits an exact opposite trend for both loss and $S_A$, where it shows the highest $S_A$ with very high loss magnitude. In kilometers-long communication fibers, only a few dB/cm loss is allowable for attaining sufficient energy output for signal reconstruction [50], however considering our optical setup (a few centimeters long fiber), the tolerable limit can be extended. Even so, this extension is not infinite by principle and that is why we had to sacrifice both $S_\lambda$ and $S_A$ to cap the loss within realizable threshold. Therefore, our selection of t = 50 nm for the conducting layer is deemed the most suitable for our desired application.

Furthermore, the proposed sensor exhibited tolerable performance fluctuations in terms of its loss characteristics while examined for dimensional erosion and dilation up to ±10% from the optimized PCF parameters. The simulation result reveals the effects of the different diameters of the air holes and the pitch size variations for analyte RI 1.3743 which are displayed in Fig 6. Modifying large air hole size does not cause any significant spectral position fluctuations. However, it regulates the peak losses to 174 dB/cm, 189 dB/cm, 218 dB/cm, and 232 dB/cm from the optimum value of 257 dB/cm when the optimized air hole size (d = 1.43 μm) was changed by +10%, + 5%, -5%, and -10%, respectively (see Fig 6a). In the case of scaled-down air holes, the sensor shows a slight change in loss characteristics again for ± 10% variations in their size. The maximum and minimum loss values of 214 dB/cm at 1645 nm and 194 dB/cm at 1626 nm were found for changes of +10% and -10% respectively from the proposed diameter (dc = 0.44 μm) which are only +5.4% and -4.4% of the optimum value (see Fig 6b). The wavelength shift of > 20 nm compensates with the higher RI analytes and does not affect the performance in any way. Additionally, Fig 6c shows the spectra of loss depth caused by varying pitch size. Loss peaks are gradually reduced when the pitch size is increased, and vice

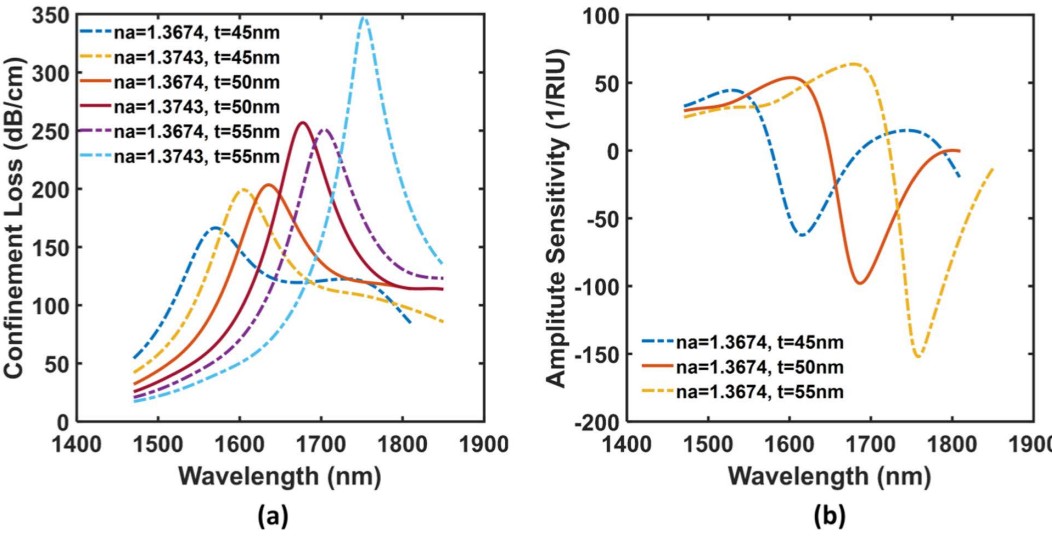

**Fig 5.  (a) Loss spectrum and (b) Amplitude Sensitivity spectrum for varying ITO thickness in Y-polarized mode.**

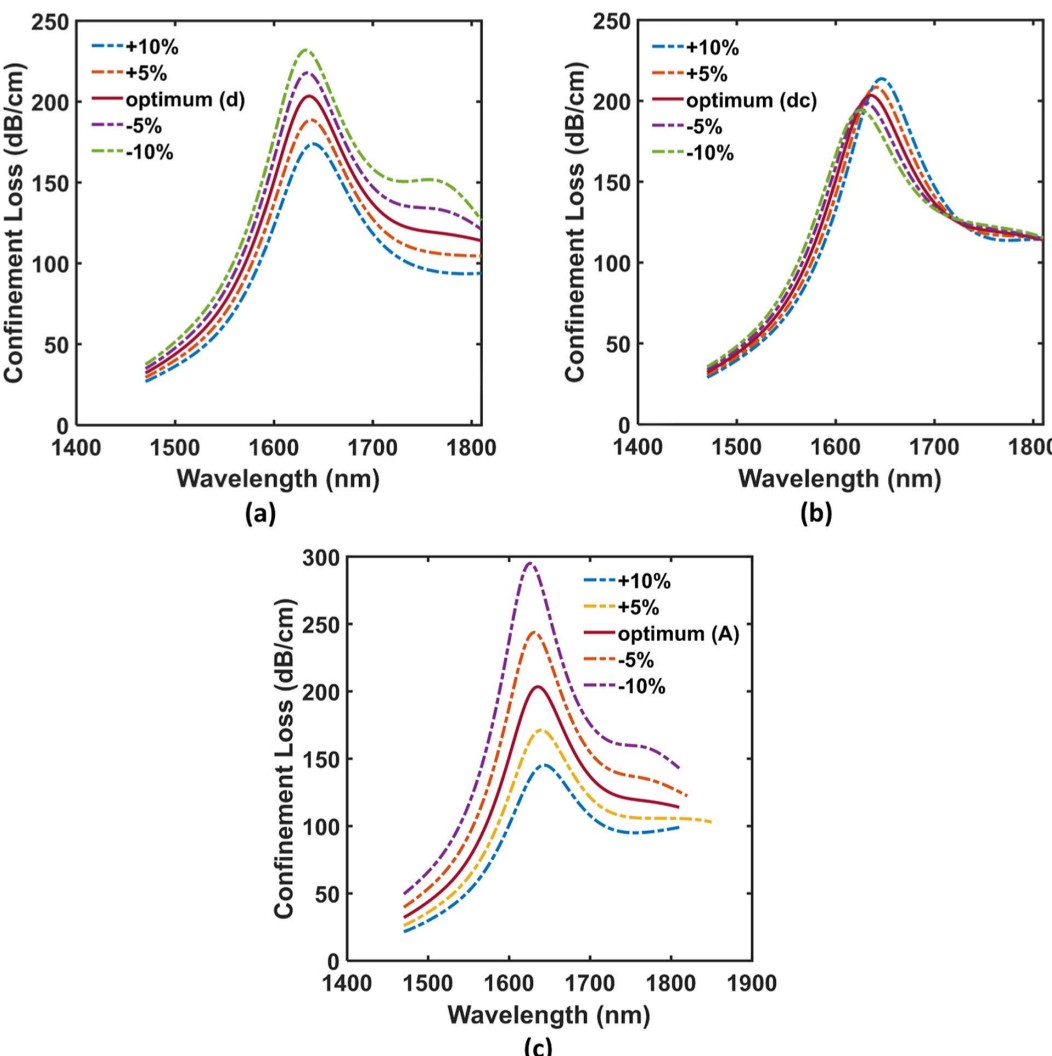

**Fig 6. Fabrication tolerance effects on (a) Large air hole variations, (b) Small air hole variations, and (c) Pitch size variations for Y-polarized mode.**

versa. In contrast, as a result of any reduction in pitch size, the RI contrast difference between core mode and SPP mode is elevated, which significantly raises the loss depths resulting in an improved FOM. According to Fig 6c, if the optimum pitch size is modified by ±10%, the sensor illustrates peak losses of 145 dB/cm, 171 dB/cm, 245 dB/cm, and 295 dB/cm, respectively. Besides, the fluctuations were extremely small (fractions of nanometer) in terms of spectral shift. Simulation-wise, it is apparent that even if we modify all the suggested variables within a range from +10% to -10%, its effect on the sensor's output remain trivial. Hence, if the loss depths and resonant peak wavelengths slightly change from their optimum values, the sensor can still ensure highly consistent operation.

## Conclusion

In summary, the proposed SPR-PCF sensor is developed for rapid serum sodium electrolyte detection that shows highly optically sensitive performance. We numerically analyzed

the detection performance of our SPR-PCF sensor to confirm the best sensing operation. The sensor resolution spans from $6.6 \times 10^{-6}$ RIU to $2.0 \times 10^{-5}$ RIU with the maximum wavelength and amplitude sensitivity of 15,160 nm/RIU and 470 RIU$^{-1}$, respectively. In the FEM-based simulations, we validated the structural tolerances are within practical limits. Additionally, it shows a high birefringence peak of approximately $11.4 \times 10^{-4}$. We anticipate that the proposed polarization-sensitive sensor will be an effective POC tool for hypo- or hypernatremia fast identification in both at-home and nosocomial circumstances.

## Supporting information

**S1 File. All data underlying the results presented in this paper and necessary to replicate our study findings are available from the COMSOL Multiphysics file (S_sens_ito.m file) that is zipped and attached as the supporting information.**
(ZIP)

## Acknowledgment

A.M.T.H and A.I authors contributed equally to this work.

## Author contributions

**Conceptualization:** Abrar Islam.

**Data curation:** Abdullah Mohammad Tanvirul Hoque.

**Formal analysis:** Abdullah Mohammad Tanvirul Hoque, Abrar Islam, Firoz Haider, Rifat Ahmmed Aoni, Rajib Ahmed.

**Investigation:** Abrar Islam.

**Software:** Abdullah Mohammad Tanvirul Hoque, Firoz Haider.

**Supervision:** Rifat Ahmmed Aoni, Rajib Ahmed.

**Validation:** Abrar Islam, Firoz Haider, Rifat Ahmmed Aoni, Rajib Ahmed.

**Visualization:** Abdullah Mohammad Tanvirul Hoque, Firoz Haider.

**Writing – original draft:** Abrar Islam.

**Writing – review & editing:** Abrar Islam, Firoz Haider, Rifat Ahmmed Aoni, Rajib Ahmed.

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
