## [Decision Letter · Decision Letter 0]

28 Nov 2024

PONE-D-24-53025Point-of-care testing of hyponatremia and hypernatremia levels: an optoplasmonic biosensing approachPLOS ONE

Dear Dr. Islam,

Thank you for submitting your manuscript to PLOS ONE. After careful consideration, we feel that it has merit but does not fully meet PLOS ONE’s publication criteria as it currently stands. Therefore, we invite you to submit a revised version of the manuscript that addresses the points raised during the review process.

We look forward to receiving your revised manuscript.

Kind regards,

Yuan-Fong Chou Chau

Academic Editor

PLOS ONE

Journal Requirements:

2. In the online submission form, you indicated that [Data underlying the results presented in this paper are available from the corresponding authors upon request.].

Reviewers' comments:

Reviewer's Responses to Questions

**Comments to the Author**

1. Is the manuscript technically sound, and do the data support the conclusions?

Reviewer #1: Yes

Reviewer #2: Partly

Reviewer #3: Partly

2. Has the statistical analysis been performed appropriately and rigorously? 

Reviewer #1: Yes

Reviewer #2: Yes

Reviewer #3: I Don't Know

3. Have the authors made all data underlying the findings in their manuscript fully available?

Reviewer #1: Yes

Reviewer #2: Yes

Reviewer #3: Yes

4. Is the manuscript presented in an intelligible fashion and written in standard English?

Reviewer #1: Yes

Reviewer #2: Yes

Reviewer #3: Yes

5. Review Comments to the Author

Reviewer #1: Comments to Author

1. How does the imbalance of Na+ ions in blood or lymph impact a patient’s health and why is it crucial to monitor sodium ion (Na+) levels in the human body properly explain the medial cause?

2. What limitations in current point-of-care (POC) testing for Na+ levels does this study aim to address and do photonic crystal fiber (PCF)-based surface plasmon resonance (SPR) sensor detect those shortcomings?

3. How does the modified square lattice design improve the waveguidance and sensing performance of the proposed PCF sensor

4. Why was ITO chosen as the plasmonically active material, over other materail like Au, Ag and how does it enhance the sensor's performance?

5. How does structural asymmetry-induced birefringence contribute to the superior sensing response in the y-polarized mode compared to the x-polarized mode?

6. What does a sensitivity of 15,157 nm/RIU indicate about the sensor's performance or this is just a numerical figure only?

7. What is the significance of achieving a detection resolution in the order of 10−6 RIU in practical applications?

8. How does the proposed sensor compare to conventional serum electrolyte panels currently used in medical diagnostics?

9. In what ways could this sensor be utilized as a POC alternative for rapid Na+ ion testing?

10. Update the introduction section of the article with some latest models of PCF SPR sensor like mereger of IMD and EMD shaped model, EMD and D shaped model, meger of spectroscopy and RI sensing etc. Cite them in the intoduction section?

https://doi.org/10.1007/s11468-023-01940-8

https://doi.org/10.1016/j.measurement.2021.110513

https://doi.org/10.1016/j.ijleo.2022.169892

Reviewer #2: The paper presents a novel photonic crystal fiber-based surface plasmon resonance (SPR) sensor for rapid, in situ point-of-care testing of sodium levels in blood, addressing the limitations of current technologies with its high sensitivity and resolution. This sensor offers a promising alternative for monitoring conditions like hyponatremia and hypernatremia, potentially improving diagnostic speed and accuracy in medical settings.

Overall, I believe that this paper is useful for the optics and biosensing community. However, I have a few comments that the authors should consider:

1. In the description under Figure 3, it is stated that “The sensor sensitivity peaks at test RI 1.3674 (concentration = 440 mg/dL) to a magnitude of 470 RIU-1 in the amplitude interrogation as visualized in the figure”. However, according to Figure 3(b), the amplitude sensitivity of 470 RIU-1 corresponds to an RI of 1.3810. Moreover, Table II lists the concentration of 440 mg/dL as correlating with an RI of 1.3874. Can the authors clarify this apparent discrepancy? It is crucial to ensure that the text accurately reflects the data presented in figures and tables.

2. The manuscript states under Table II that “Initially, a polarizer followed by a broadband light source inputs the IR wavelengths into the fiber over a preassigned interval (typically a few seconds).” This sequence suggests that the polarizer is positioned before the light source, which is unconventional and possibly incorrect. Typically, the light source should precede the polarizer in the setup. Therefore, the correct phrasing should likely be, “A broadband light source followed by a polarizer inputs the IR wavelengths into the fiber…” Additionally, to ensure alignment with this setup, it is recommended to include a polarizer oriented in the Y-direction between the laser source and the single-mode fiber (SMF) in Figure 1(c). Could the authors revise the experimental setup accordingly to avoid potential confusion?

Reviewer #3: The authors propose a photonic crystal fiber (PCF) sensor with a modified square lattice to achieve a propagation-controlled core, enhancing wave guidance and improving sensing performance. While the manuscript aligns with the journal's scope and presents intriguing findings, clarification of underlying mechanisms and additional references are required. Major revisions are recommended before acceptance.

Comments for Revision:

1. In the introduction, briefly elaborate on the novelty of this work and clarify the unique properties of the proposed SPR PCF sensor structures compared to other SPR sensor approaches (e.g., Nanomaterials 11(8), 2097 (2021); Micromachines 14(2), 340 (2023), Plasmonics 19 (1), 481-493 (2024) and DOI: https://doi.org/10.1007/s11468-024-02319-z). Include these references to strengthen the context.

2. Provide detailed settings for the COMSOL simulations, including the thickness of the perfectly matched layer (PML), scattering boundary conditions, and mesh size.

3. Specify the values of Sellmeier constants B1, B2, B3, C1, C2, and C3 in the manuscript.

4. Add relevant references on SPR PCF sensors, such as Crystals 13(5), 813 (2023) and Photonics 9(12), 916 (2022), to enhance the mechanism and background of SPR PCF sensors in context.

5. Define birefringence in the text and include references on birefringence in PCFs, such as Prog. Electromagn. Res. B 22, 39–52 (2010) and Jpn. J. Appl. 46(11L), L1048 (2007).

6. Improve the quality of Figures 2(a) and (b) and include a color scale bar for better visualization.

7. Add the simulation method and emphasize the potential applications of this work in the conclusion section.

8. For Fig. 3(a), explain the mechanism behind the significant change in confinement loss when nₐ varies from 1.3743 to 1.3810.

9. Additionally, clarify why nₐ = 1.3874 exhibits a larger FWHM compared to lower nₐ values.

10. Finally, specify the detection range of nₐ for the designed SPR PCF sensor.

6. PLOS authors have the option to publish the peer review history of their article (what does this mean? ). If published, this will include your full peer review and any attached files.

**Do you want your identity to be public for this peer review?** For information about this choice, including consent withdrawal, please see our Privacy Policy .

Reviewer #1: No

Reviewer #2: No

Reviewer #3: No

---

## [Author Response · Author response to Decision Letter 1]

3 Feb 2025

Please find the point-by-point responses to the reviews comments in the attached file titled "Response to Reviewers".

---

## [Editor Report · Decision Letter 1]

5 Feb 2025

Point-of-care testing of hyponatremia and hypernatremia levels: an optoplasmonic biosensing approach

PONE-D-24-53025R1

Dear Dr. Islam,

We’re pleased to inform you that your manuscript has been judged scientifically suitable for publication and will be formally accepted for publication once it meets all outstanding technical requirements.

Kind regards,

Yuan-Fong Chou Chau

Academic Editor

PLOS ONE
---

## [Editor Report · Acceptance letter]

PONE-D-24-53025R1

PLOS ONE

Dear Dr. Islam,

I'm pleased to inform you that your manuscript has been deemed suitable for publication in PLOS ONE. Congratulations! Your manuscript is now being handed over to our production team.

Kind regards,

on behalf of

Dr. Yuan-Fong Chou Chau

Academic Editor

PLOS ONE
